# The Role of Protein Methyltransferases in Immunity

**DOI:** 10.3390/molecules29020360

**Published:** 2024-01-11

**Authors:** Chaoran Song, Mi-Yeon Kim, Jae Youl Cho

**Affiliations:** 1Department of Integrative Biotechnology, Sungkyunkwan University, Suwon 16419, Republic of Korea; songchaoran115@163.com; 2School of Systems Biomedical Science, Soongsil University, Seoul 06978, Republic of Korea

**Keywords:** protein methylation, inflammation, arginine methylation, lysine methylation

## Abstract

The immune system protects our body from bacteria, viruses, and toxins and removes malignant cells. Activation of immune cells requires the onset of a network of important signaling proteins. Methylation of these proteins affects their structure and biological function. Under stimulation, T cells, B cells, and other immune cells undergo activation, development, proliferation, differentiation, and manufacture of cytokines and antibodies. Methyltransferases alter the above processes and lead to diverse outcomes depending on the degree and type of methylation. In the previous two decades, methyltransferases have been reported to mediate a great variety of immune stages. Elucidating the roles of methylation in immunity not only contributes to understanding the immune mechanism but is helpful in the development of new immunotherapeutic strategies. Hence, we review herein the studies on methylation in immunity, aiming to provide ideas for new approaches.

## 1. Introduction

Methylation is a post-translational modification (PTM) associated with a variety of cellular functions through enzymatic modification of proteins. Transmethylation is orchestrated by writers (e.g., methyltransferases), readers (e.g., binding substrate proteins), and erasers (e.g., demethylases) with distinct roles in adding, recognizing, or removing these methyl groups. Methyltransferases transfer methyl groups from a donor, generally S-adenosyl-L-methionine (AdoMet), to different acceptor molecules [1]. At present, the AdoMet-dependent methyltransferases have been divided into three families [2]. The most abundant group (Class I) contains a seven-strand twisted β-sheet structure [3]. The methyltransferases in the second group (Class II) possess a conserved Su(var)3-9, enhancer-of-zeste and Trithorax (SET) domain structure, which is approximately 130 amino acids long [4]. Class III consists of methyltransferases that are enzymes with multiple membrane-spanning regions [5]. In eukaryotes, arginine and lysine are multiply methylated and lead to distinct outcomes. Methylation of arginine and lysine provides significant functional diversity and regulatory complexity [4]. 

Protein arginine methyltransferases (PRMTs), a group of methyltransferases with a seven β-strand set, methylate proteins on arginine residues [6]. Arginine-mediated methylation is the most prevalent type of protein methylation in mammalian cells. It is involved in signal transduction, RNA processing, chromatin stability, and transcriptional regulation [6,7,8]. The AdoMet-dependent methyltransferases in the PRMT family share four conserved motifs (I, post-I, II, and III), as well as a THW loop [9]. The AdoMet-binding pocket mainly consists of Motifs I, post-I, and a THW loop [9]. These motifs are highly conserved in eukaryotes, particularly in a core region that contains ~310 amino acids responsible for catalyzing the enzymatic activity of the group [10,11,12]. Three types of arginine methylation have been identified (Figure 1). PRMTs transfer a methyl group from AdoMet to a guanidino group of arginine, leading to monomethylarginine and asymmetric dimethylarginine (ADMA) in target proteins [13]. The addition of two methyl groups to the two *ω*-guanidino nitrogen atoms of arginine forms symmetrically dimethylated arginine (SDMA) [14]. The structural domain of PRMTs is shown in Figure 2.

Lysine methyltransferases (KMTs) add one, two, or three methyl groups from the donor, generally *S*-adenosyl-l-methionine, to the ε-nitrogen of lysine, undergoing mono-, di-, and trimethylation (me1, me2, and me3, respectively) (Figure 3) [15,16]. The majority of known KMTs contain a conserved SET domain, enabling them to bind with SAM [17,18]. Although numerous KMTs were initially regarded as histone methyltransferases, accumulating methyltransferases have been identified to target histones and non-histones. Several KMTs have been identified to be specific for non-histones [19,20,21]. Thus, these methyltransferases are endowed with distinctive functions in cellular processes, transcriptional activation, and repression. Demethylases allow reversible conversion of methylation by removing a methyl group from their substrates [22]. Abnormal methylation has been reported to be associated with several diseases, including cancer, acute myeloid leukemia, Parkinson’s disease, and immunopathies [23,24,25]. Therefore, methyltransferases and their binding proteins can be developed as therapeutic targets [26]. In this review, we summarize the role of methyltransferases in immunity. The structural domain of PRMTs is shown in Figure 4.

## 2. The Role of Arginine Methyltransferases in Inflammatory Responses

To date, the mammalian family of PRMTs is recognized to have nine members. They are grouped into two categories according to the type of modification they catalyze [27]. Type I enzymes are PRMTs that add two methyl groups to the same terminal nitrogen group of arginine and form ADMA. PRMT1, -2, -3, -4, -6, and -8 belong to this type [10,28]. PRMT5, -7, and -9 are type II enzymes that transfer a second methyl group to the other terminal nitrogen, generating SDMA. In Table 1, immunopathological responses, target molecules, and molecular outcomes of protein arginine methyltransferases are summarized.

### 2.1. PRMT1 

As the first identified PRMT in mammals, PRMT1 is involved in up to 85% of total arginine methylation activity [56,57]. In mammals, the NF-κB family is composed of five transcription factors: p50, p52, RelA (p65), c-Rel, and RelB [58,59]. The classical NF-κB pathway is essential for innate immunity through IKK-dependent IκB degradation (Figure 5) [60]. Additionally, the alternative pathway of NF-κB plays a distinct role in lymphoid organ development and adaptive immunity [60]. 

PRMT1 is considered a regulator of immunity based on its effect on NF-κB. PRMT1 directly binds to TRAF6 and reversibly methylates it at multiple sites in both primary and cultured cells (Figure 6). After being arginine-methylated by PRMT1, the ubiquitin ligase activity of TRAF6 is downregulated, leading to the suppression of basal NF-κB activation. The loss of PRMT1 enhances the activities of IRAK3 and NF-κB reporter luciferase as well as the nuclear translocation of p65 [29]. These data suggest that PRMT1 is essential for blocking TRAF6-dependent pathways, and that the absence of PRMT1 results in impaired Toll-like receptor (TLR) ligand constitutive activation and response [29]. Hassa et al. have shown direct interaction between PRMT1 and NF-κB subunit p65. The interaction activates NF-κB-dependent gene expression at the promoters of CARM1-dependent macrophage inflammatory protein 2 (MIP2) and human immunodeficiency virus 1 (HIV-1) [30]. Luciferase reporter gene assays have strongly indicated that PRMT1 has synergy with poly(ADP-Ribose) polymerase 1 (PARP1) and p300 [30]. Interestingly, the knockdown of PRMT1 increases the levels of NF-κB target genes by facilitating p65 recruitment to their promoters in response to TNFα. PRMT1 is a restrictive factor of NF-κB by interacting with p65 [31]. 

Weber et al. have demonstrated in both in vitro and in vivo tests that PRMT1 also methylates the protein inhibitor of activated STAT1 (PIAS1), a negative regulator of STAT1 in immune responses [32]. Arginine methylation of PIAS1 is vital for the inhibition of PRMT1 in interferon (IFN)-dependent transcription. The methylation also recruits PIAS1 to STAT1 target promoters in response to IFN [33]. Further previously reported evidence of the significance of methylation was that STAT1 is methylated by PRMT1 on Arg-31 and that methylation of STAT1 is required for transcriptional activation. Treatment with methylthioadenosine, a transmethylation inhibitor, weakens DNA–STAT1 binding by strengthening the connection between STAT1 and PIAS1. The loss of PRMT1 decreases the anti-viral and anti-proliferative capacities of type 1 IFNs, showing the role of PRMT1 in IFNα/β-receptor-associated signaling events [34]. A two-hybrid screening system has indicated that PRMT1 is the first enzyme in its family to bind to the cytoplasmic region of type 1 IFN receptor [61,62]. PRMT1 regulates peroxisome proliferator-activated receptor γ (PPARγ) gene expression at its promoter through H4R3me2a methylation, leading to impaired expression of PPARγ in response to interleukin-4 (IL-4) treatment. Using a myeloid-specific PRMT1 knockout mouse model, Tikhanovich et al. showed that the abolition of PRMT1 causes a lower survival rate and higher pro-inflammatory cytokine production after cecal ligation and puncture [35]. These results indicate that the regulation of PPAR in PRMT1-dependent macrophages can cause PRMT1 knockout mice to be susceptible to infection.

Furthermore, PRMT1 activity is crucial for proper execution of several processes that are important for humoral immunity in B cells [36]. The expression and activity of PRMT1 in human and mouse peripheral blood B cells increase after activation in vitro or in vivo. A marked decrease in the immune system response to T cell-dependent antigens has been observed when PRMT1 is deleted. Upon stimulation of multiple mitogens, activation of PRMT1-impaired B cells results in reduced survival, proliferation, and differentiation in vitro. 

Additionally, PRMT1 alters the differentiation, generation, and activation of T cells (Figure 7). PRMT1 is located downstream of the T cell receptor in progenitor Th cells [40]. T cell receptor signaling promotes the expression of PRMT1 (https://www.sciencedirect.com/topics/biochemistry-genetics-and-molecular-biology/prmt1, accessed on 22 December 2023), which methylates the arginine residues of NIP45, the cofactor protein of nuclear factor of activated T cells [40]. Methylation facilitates its association with NFAT and leads to elevated cytokine production, whereas deficient expression of IFN-γ and IL-4 has been observed in T cells from NIP45-impaired mice [37]. 

Retinoic acid-related orphan receptor γt (RORγt) is a key transcription factor that mediates the differentiation of Th17 cells [38,63]. PRMT1 has been demonstrated to be related to RORγt and regulated mouse Th17 differentiation, which is promoted by PRMT1 overexpression. However, the knockdown of PRMT1 by shRNA and inactivation by specific PRMT1 inhibitors limits Th17 differentiation. The use of a specific inhibitor of PRMT1 damages the production of Th17 cells and alleviates activation of experimental autoimmune encephalomyelitis in mice [38]. Sen et al. have suggested that PRMT1 could be a new target for reducing Th17-mediated autoimmunity by decreasing the generation of pathogenic Th17 cells [38]. Vav1, a Rac/Rho guanine nucleotide exchange factor, plays a crucial role in cytokine secretion, T cell activation, and proliferation [64,65]. Methylation of Vav1 is promoted in both human and mouse T cells and occurs in the nucleus [39,40]. The inhibition of cellular transmethylation of PRMT1 reduces methylation of Vav1 and IL-2 production, indicating potentially crucial roles for PRMT1 in T cell-mediated disorders.

### 2.2. CARM1

CARM1 (PRMT4) is a transcriptional coactivator related to the p160 family in nuclear receptor-mediated transcription [66]. In addition to the p160 family, CARM1 synergistically activates NF-κB-mediated transactivation with P300/CREB-binding protein [41,67,68]. As reported, CARM1 bound to p300 in vivo and interacted with p65, a NF-κB subunit, in vitro [43]. During TNFα or LPS stimulation, CARM1^−/−^ mouse embryonic fibroblasts exhibited dampened expression of a group of NF-κB target genes [43,69]. The RNA-binding protein HuR, a novel substrate of CARM1, is methylated by CARM1 at arginine 217 [70]. Methylation of endogenous HuR and stabilization of TNF-α mRNA have been observed in lipopolysaccharide-stimulated macrophages. The methylation of HuR was similarly increased in cells overexpressing CARM1, and methylated HuR is related to the stability of HuR-dependent mRNA [44]. Previous research has shown that CARM1 is associated with survival of thymocytes. Thymocytes isolated from CARM1-deleted embryos are stagnated between CD4^−^ CD8^−^ double-negative stage 1 and double-negative stage 2 [45,66]. A significant reduction in the number of thymocytes has been observed. Therefore, the methylation of arginine residues by CARM1 in inflammation suggests CARM1 as a therapeutic target for inflammatory diseases [66].

### 2.3. PRMT5

Acute graft-versus-host disease (aGVHD) is a T cell-mediated immune dysfunction in which T cells in donated tissue recognize the recipient as foreign [66]. PRMT5 has been identified to play a role in T cell responses in aGVHD, suggesting it as a target of this disease [46]. PRMT5 is a mediator in experimental autoimmune encephalomyelitis (EAE), a well-developed animal model of autoimmune disease multiple sclerosis [71]. In vivo EAE mouse models have shown that PRMT5 inhibition potently repressed memory T cell responses. Delayed-type hypersensitivity and inflammation in clinical disease were also decreased. The inhibition of PRMT5 by specific inhibitors downregulates IL-2 production and proliferation of recall Th cells. These results demonstrate the importance of PRMT5 as a regulator in adaptive memory Th cell responses [47]. In lymphoma cells, deceased PRMT5 represses TP53K372 methylation, cyclin D1 transcriptional activation, and BCL3 production and promotes NF-κB p52–HDAC1 repressor complexes to the cyclin D1 promoter [48]. 

As Nagai et al. have reported, PRMT5 forms a complex with FOXP3 homomer in Tregs [49]. Therefore, a specific blockade of PRMT5 decreases methylation of FOXP3 and arrests human Treg functions. Mice with conditional knockout of PRMT5 expression in Tregs develop severe scurfy-like autoimmunity and display a limited suppressive function. This may also explain the reduced numbers of Tregs in the spleen in PRMT5 cKO mice [49]. Additionally, PRMT5 has been found to regulate T cell survival and proliferation through analysis of T cell-specific PRMT5 conditional knockout mice [50]. PRMT5 is essential for natural killer T cell development, optimal peripheral T cell maintenance, and early T cell development [50]. Consistently, deficient IL-7-dependent survival and TCR-induced proliferation in T cells were caused by deletion of PRMT5 in vitro [50]. Separately, PRMT5 has been shown to be important for antibody responses and plays essential but distinct roles in all proliferative B cell stages in mice [51]. PRMT5 prevents p53-mediated suppression in pro-B and pre-B cells and inhibits apoptosis of mature B cells during simultaneous activation via p53-independent pathways [51]. The inhibition of PRMT5 markedly decreases phosphorylation of STAT1 and transcription of pro-inflammatory genes, including IL-17 and IFN-γ. Additionally, PRMT5 inhibition disrupts signaling transduction by regulating phosphorylation of ERK1/2, which leads to dysregulation of the cell cycle in activated T cells [46]. The data above indicate PRMT5 inhibitors as a novel method to treat T cell-dependent inflammatory disease.

### 2.4. PRMT6

Utilizing proteomics-based methods, a protein–protein interaction has been discerned between PRMT6 and interleukin-enhancer binding protein 2 (ILF2). Moreover, macrophage migration inhibitory factor has been shown to play a role in mediating alternative activation of tumor-associated macrophages. Avasarala et al. have identified the macrophage migration inhibitory factor as an important downstream molecule of PRMT6–ILF2 signaling [52]. HIV-1 Tat protein is a key player in HIV replication by increasing gene transcription efficiency. HIV-1 is a specific substrate of PRMT6 in vivo and in vitro that targets Tat R52 and R53 residues for arginine methylation [53]. The overexpression of PRMT6 decreases the level of Tat transactivation of HIV-1 long terminal repeat chloramphenicol acetyltransferase and luciferase reporter plasmids in a dose-dependent manner, while the knockdown of PRMT6 enhances HIV-1 production and the speed of viral replication [53]. Thus, PRMT6 disrupts the transcriptional activation of Tat and may represent an innate cellular immune form of HIV-1 replication.

PRMT6 also plays a role in immunity by targeting a series of signaling pathways. PRMT6 has been identified as an NF-κB coactivator because it can generate transgenic mice that express PRMT6 fused to the hormone-binding portion of the estrogen receptor [54]. PRMT6 engages in a direct interaction with RelA, whereby its overexpression amplifies the transcriptional activity of an ectopic NF-κB reporter and intrinsically regulates NF-κB target genes [54]. In response to TNF-α stimulation, RelA recruits PRMT6 to specific NF-κB target promoters. Phosphatase and tensin homolog (PTEN) is recognized as a tumor-suppressor gene, and its mutation has implications in the progression of various cancers [70]. PRMT6 interacts with PTEN and methylated PTEN R159, weakening the PI3K–AKT cascade [72]. G protein pathway suppressor 2 (GPS2) cytoplasmic actions and anti-inflammatory roles are linked with the regulation of JNK activation as well as TNF-α target genes in macrophages [73]. Interaction with the exchange factor TBL1 is helpful to protect GPS2 from degradation. The methylation of GPS2 by PRMT6 modulates the interaction with TBL1 and suppresses proteasome-dependent degradation [74]. PRMT6 also attenuates anti-viral innate immunity by blocking TBK1–IRF3 signaling [55]. In PRMT6-deficient mice, the TBK1–IRF3 interaction is enhanced and activates IRF3 as well as increases the production of type I IFN. A Viral infection not only upregulates PRMT6 protein levels, but also promotes the binding between PRMT6 and IRF3 and dampens the interaction between IRF3 and TBK1 [55].

## 3. The Role of Lysine Methyltransferases in Inflammatory Responses

PKMTs predominantly belong to the class V superfamily, containing a conserved SET domain. Methyltransferases with SET domains stand out from other AdoMet-dependent methyltransferases due to their distinctive binding sites for substrates and AdoMet. The immunopathological responses, target molecules, and molecular outcomes of protein lysine methyltransferases are summarized in Table 2.

### 3.1. G9a 

In innate immune cells such as macrophages, G9a-dependent H3K9me2 can be involved in gene repression during endotoxin tolerance [107]. Stimulated by LPS, a novel chromatin activity-based chemoproteomic (ChaC) approach has revealed that macrophages acquire endotoxin tolerance to further LPS stimulation through the activation of 9a-dependent H3K9me2 [108]. In tolerized macrophages, G9a interacts with ATF7, a member of the ATF-CREB subfamily, as well as heterochromatin protein 1 (HP1) and RelB, members of the NF-κB family [75,109,110]. These factors recruit G9a to a specific silenced promoter and lead to gene silencing during cellular responses to inflammatory signals [107]. Consistent with those findings, Merkling et al. have demonstrated G9a-regulated tolerance to virus infection in *Drosophila* by limiting JAK/STAT signaling [76]. Due to hyperactivation of JAK/STAT and its target genes, G9a-deficient flies succumb faster to infection with RNA viruses than control flies [76]. Therefore, G9a is a key mediator of gene expression in innate inflammatory processes. 

Previous studies have indicated that G9a-mediated H3K9Me2 most likely plays a role in the transient transcriptional repression of genes associated with lymphocytes or lymphocyte-specific functions [107]. ChIP-on-chip experiments have identified several H3K9Me2-enriched pathways in lymphocytes including T cell receptor signaling, IL-4 signaling, and GATA3 transcription [107,111]. A G9a knockout strain has been generated (G9a^fl/fl^) to elucidate the immune function of G9a in the lymphatic system [77]. In agreement with a previous study [112], G9a has been shown to be required for Th cell differentiation rather than T cell homeostasis or antigen-presenting cell function. The numbers of CD4^+^ and CD8^+^ T cells are comparable in the thymus and spleen of both G9a^fl/fl^ and G9a^−/−^ mice. Moreover, bone marrow-derived dendritic cells from both G9a^fl/fl^ and G9a^−/−^ mice are equally able to promote antigen-specific proliferation of Th cells and display normal inflammatory responses. However, downregulated levels of H3K9me2 at the IFN-γ and IL-13–IL-4 loci have been observed in G9a deficiency via a chromatin immunoprecipitation assay. During Th cell differentiation, the H3K9me2 level is dynamically upregulated, whereas the levels are decreased at lineage-specific loci. G9a-deficient Th cells fail to produce Th2-specific cytokines IL-4, IL-5, and IL-13, while G9a is dispensable for Th1 cell responses [77]. Working with a *Trichuris muris*-infected G9a ^ΔT^ mouse model, Lehnertz et al. have shown that mesenteric LN cells produce reduced expression of protective CD4^+^ T cell-derived IL-4 [77]. Taken together, these results reveal that G9a has a critical role in regulating expression of Th2-associated cytokine genes both in vivo and in vitro. 

In contrast, G9a limits Th17 and Treg cell differentiation during murine intestinal inflammation in a methyltransferase-activity manner (Figure 7) [78]. After activation under Treg or Th17-promoting conditions, G9a^−/−^ T cells exhibit an obvious increased frequency of IL-17A-producing and Foxp3^+^ Tregs. In WT T cells, a similar result is obtained by using two G9a methyltransferase activity-specific inhibitors: UNC0638 and BIX01294. Mice that receive G9a^−/−^ T cells do not display any detectable colonic inflammation and systemic disease, such as colonic shortening and splenomegaly [78]. G9a-deficient T cells lead to a notable promotion in the frequencies of both Treg and Th17 cells during intestinal inflammation, indicating that G9a negatively regulates the differentiation of these cell lineages in vivo [78]. Furthermore, G9a-dependent H3K9me2 has been observed at high levels at the IL17a, Il17f, Rorc, and FOXP3 promoters in naive Th cells and dramatically decreases after T cell activation under either Th17 or Treg-promoting conditions [78]. G9a also restricts in vitro Treg differentiation that is dependent upon its methyltransferase activity. After Treg differentiation, the H3K9me2 level in G9a^fl/fl^ T cells isolated from G9a^fl/fl^ mice is reduced to that similar to those of naive G9a^−/−^T cells isolated from G9a^−/−^ mice. Under Th17 and Treg-promoting conditions, G9a^−/−^ T cells exhibit 40-fold greater sensitivity to TGF-β than WT cells. Formaldehyde-assisted isolation of regulatory elements (FAIRE-Seq), a genome-wide sequencing technique, shows an increase in chromatin accessibility that leads to enhanced activation of the FOXP3 locus, resulting in increased frequencies of Tregs. Taken together, these results indicate G9a-dependent H3K9me2-regulated chromatin accessibility and TGF-β1 responsiveness. 

Unlike native T cells, which are recruited by a transcriptional repressor (Blimp-1), G9a aggregates to the Il2rα and CD27 loci, repressing their expression [79]. During acute viral infection, Blimp-1-deficient CD8^+^ T cells show sustained expression of CD25 and CD27 as well as persistent cytokine responses, leading to enhanced proliferation and survival. In addition to T cells, G9a regulates the development and function of innate lymphoid cells, a family of immune cells that mirror the phenotypes of T cells [113]. Mice with a hematopoietic cell-specific deletion of G9a (Vav.G9a^−/−^ mice) have an obvious reduction in group 2 innate lymphoid cells (ILC2s) in peripheral sites and a deficient development of immature ILC2s in the bone marrow [80]. Additionally, Vav.G9a^−/−^ mice have greater resistance to allergic lung inflammation [80]. Consistently, in vitro-expanded immature ILC2s from Vav. G9a^−/−^ mice produced lower levels of ILC2-specific genes including IL5 and IL13. A genome-wide expression analysis has identified a global shift in expression from ILC2-specific transcripts to group 3 innate lymphoid cell (ILC3)-specific genes, placing G9a and H3K9me2 as central regulators of the ILC2/ILC3 lineage choice [80,107]. Taken together, these results suggest that G9a plays an essential role in immune cell function. 

### 3.2. EHMT1

It has been reported that EHMT1 interacts with NF-κB p50 and suppresses gene expression [81]. EHMT1 promotes H3K9 methylation at the promoters of NF-κB-regulated genes such as IL-8. Furthermore, p50 employs EHMT1 to the promoters of type I IFN-responding genes through the interaction between EHMT1 and RHD-N, one of the functional domains of p50 [81]. EHMT1- and p50-impaired cells are resistant to viral infection using RNA interference. Mono-methylation of RelA by SETD6 at K310 leads to the suppression of NF-κB signaling by docking EHMT1 at target genes to generate a silent chromatin state [82]. EHMT1 has also been indicated to suppress differentiation of induced regulatory T cells. The combined approach of DNA pull-down and mass spectrometric analysis has revealed that Wiz, a transcription factor, recruits EHMT1 to FOXP3 Treg-specific demethylated regions (TSDRs) [83]. During iTreg differentiation, the knockout of EHMT1 or Wiz reduces H3K9me2 expression while increasing FOXP3 expression. Additionally, IFI16, an innate immune DNA sensor, activates the inflammasome and produces pro-inflammatory cytokines [114]. During Kaposi’s sarcoma-associated herpesvirus (KSHV) infection, IFI16 interacts with EHMT1 and recruits it to the KSHV genome, causing the deposition of H3K9me2/me3 and silencing of KSHV lytic genes [84]. 

### 3.3. SETDB1

Hachiya et al. have discovered that SETDB1 in macrophages effectively blocks TLR4-mediated expression of pro-inflammatory cytokines in a methyltransferase-activity-dependent manner both in vivo and in vitro [85]. SETDB1 inhibits the transcriptional activity of IL-6 promoter through H3K9 methyltransferase activity, whereas SETDB1 deficiency augments NF-κB p65 recruitment to the IL-6 promoter. However, SETDB1-deficient mice are susceptible to LPS challenge. Upon LPS stimulation, macrophage-specific SETDB1-knockout mice possess an elevated serum IL-6 level and are more sensitive to endotoxin shock. In addition, OX40, a member of the tumor necrosis factor receptor superfamily, represses IL-17 expression by activation of the NF-κB family members RelB and SETDB1 [86,87]. OX40 requires RelB to recruit SETDB1 to arrest IL-17 induction in vivo and in vitro by forming a “closed” chromatin structure at the IL-17 locus. Furthermore, a bioinformatic analysis suggests that SETDB1-related differentially expressed genes are mainly enriched in antigen processing and presentation, as well as in immune networks in breast cancer [115].

SETDB1 has been demonstrated to regulate T cell function. SMAD3 recruits SETDB1 to the proximal region of IL-2 promoter and represses IL-2 transcription in primary T cells [107]. Martin et al. have generated a T cell-specific conditional knockout of SETDB1 [88] that decreases the number of CD69^+^ and T-cell receptor TCRβ^+^ cells and promotes apoptosis in the double-positive compartment. SETDB1 interacts with the FcγRIIb promoter, and its H3K9 tri-methylation is dependent on the expression of SETDB1. The recovery of FcγRIIb due to SETDB1 deletion leads to enhanced signaling through the TCR complex. Other researchers have reported that SETDB1 is critical for T cell development due to FcγRIIb [89]. The thymocyte-specific deletion of SETDB1 impairs T cell development, especially CD8 lineage cells. SETDB1^−/−^ thymocytes show hyperexpression of FcγRIIb, leading to defective TCR-induced ERK activation. Consistent with these findings, suppressed ERK activity is restored by genetic depletion of FcγRIIb in SETDB1^−/−^ thymocytes. Adoue et al. have found that SETDB1 regulates T helper cell lineage integrity by suppressing endogenous retroviruses [116]. Th1 priming is promoted in SETDB1^−/−^-naive CD4^+^ T cells. SETDB1 deposits a repressive H3K9me3 mark at a limited and cell-type-specific set of endogenous retroviruses that are located in neighboring genes involved in immune processes, inhibiting the Th1 signature gene program in Th2 cells [116].

In B cells, SETDB1 maintains proper repression of retrotransposon sequences and regulates B cell development [90]. The deletion of SETDB1 in pro-B cells completely abolishes B cell development. In pro-B cells, the reactivation of endogenous leukemia virus causes the activation of the unfolded protein response pathway and apoptosis. Also, SETDB1 suppresses endogenous and exogenous retroviruses in B lymphocytes [91].

Additionally, Cuellar et al. have identified SETDB1 as a new negative regulator of innate immunity through a CRI SPR/Cas9 genetic screen [92]. The deletion of SETDB1 in acute myeloid leukemia human cell lines cause desilencing of retrotransposable elements that result in the induction of viral response genes. The disruption of SETDB1 reduces H3K9me3 at repetitive loci and enhances the production of double-stranded RNA [92]. The genetic ablation of SETDB1 induces a type I IFN response and an apoptosis in multiple cancer cell lines. Taken together, these results indicate that SETDB1 plays a crucial role in immunity.

### 3.4. SMYDs

An ever-increasing number of reports have shown that the SMYD protein family plays an critical role in immunity [117]. SET and MYND domain-containing 2 (SMYD2), an H3K36-specific methyltransferase, is a negative regulator of macrophage activation [93]. In activated macrophages, overexpressed SMYD2 damages IL-6 and TNF-a production as well as NF-κB and ERK signal pathways through H3K36 dimethylation. Furthermore, a high expression of macrophages of SMYD2 reduces Th-17 cells but promotes Treg cell differentiation. TGFβ induced SMYD3, a H3K4 methyltransferase, in iTreg differentiating cells [94]. SMYD3 promotes FOXP3 expression in iTreg cells. Impaired SMYD3 causes decreased H3K4me3 in the promoter region. SMYD3^−/−^ mice demonstrate increased severity of pulmonary viral infection [94]. SMYD5 is a methyltransferase that specifically trimethylates H4K20 on a subset of pro-inflammatory promoters [95]. Upon stimulation of LPS and Kdo2 lipid A, siRNAs directed against SMYD5 lead to elevated transcriptional responses of a group of TLR4 target genes, including Ccl4, IL1α, and Cxcl10 [95]. These data suggest that SMYD5 is a negative regulator of inflammatory response genes. 

### 3.5. EZH2

Enhancer of zeste homolog 2 (EZH2) is a histone methyl transferase subunit of a polycomb repressor complex that methylates H3K27 [118]. EZH2 plays a key role in T cell development, differentiation, and plasticity. Jacobsen et al. have found that EZH2 is critical for the development of adaptive instead of innate lymphoid cells using a murine model of lymphoid EZH2 deficiency [96]. The deletion of EZH2 limits H3K27me3 in thymic progenitors and promotes Cdkn2a, a cyclin-dependent inhibitor. The double deficiency of Cdkn2a and EZH2 recovers lymphocyte survival and differentiation. In the presence of Th0, Th1, and Th2, EZH2 deletion promotes the generation of IFN-γ and IL-10 [96]. Similarly, the deficiency of EZH2 in CD4^+^ T cells has been demonstrated to be associated with an elevated expression of IFN-γ, IL-13, and IL-17 in Th1, Th2, and Th17 conditions, respectively [97,107]. Coincident with the above results, EZH2-deficient Th cells have been shown to generate significantly more IL-4, IL5, IL-13, and IFN-γ during stimulation [98]. The loss of EZH2 in T cells results in enhanced production of T-bet and Gata-3, transcription factors that control T cell differentiation, and impairs H3K27me3 at their loci [99]. The absence of EZH2 in Tregs reduces the production of FoxP3, BACH2, and Neuropilin-1, which are essential genes for Treg stability [119,120,121,122]. The deficiency of EZH2 in CD4 T cells leads to a decreased number of Treg cells and differentiation but enhances the production of memory CD4 T cells [100]. Zhang et al. have found that the loss of EZH2 induces apoptosis of effector T cells through MLKL, FasL, and TNFR1 signaling [101]. Collectively, EZH2 has been suggested as a suppressor in the development and differentiation of Th1 and Th2. 

Pro-B cells and pre-B cells exhibit a higher expression of EZH2 compared to immature and recirculating B cells. The frequencies and numbers of pre-B cells and immature B cells are reduced in EZH2-deleted bone marrow cells in vivo. The frequencies of µ chain-expressing cells are three times that of a control group versus EZH2^−/−^ pro-B cells [102]. The germinal center (GC) is an inducible lymphoid microenvironment that enables antigen-activated B cells to manufacture high-specific antibodies and recall B cells [123]. The deficiency of EZH2 obviously decreases the number of splenic GC B cells in vitro as well as downregulates the number and size of GCs in EZH2^−/−^ compared to Ezh2^+/+^ mice [103]. The loss of EZH2 also leads to defective affinity maturation of immunoglobulin and the formation of high-affinity antibodies. Similarly, specific inactivation by GSK503, an EZH2 methyltransferase inhibitor, suppresses the number and volume of GC B cells and decreases high-affinity antibodies [103]. The expression of H3K27me3 in splenocytes is inhibited by GSK503 in vivo. Taken together, these findings demonstrate that EZH2 represses the formation and function of GC B cells through its methyltransferase activity. Moreover, EPZ-6438, a catalytic activity inhibitor of EZH2, decreases the proliferation of pre-B and B cells. The inhibition of EZH2 induces B cells to undergo plasma cell transcriptional changes that lead to the maturation of plasma cells and immunoglobulin secretion [104]. 

Previous studies have also shown EZH2 to mediate macrophage activation. IFN-γ recruits EZH2 and accumulates H3K27me3 at the promoters of several genes, including RANK, MERTK, and PPARG. Thus, the anti-inflammatory function of these genes is suppressed beyond the end of IFN-γ signaling [124]. The loss of EZH2 in peripheral macrophages attenuates autoimmune inflammation in a mouse model of colitis [105]. The repression of EZH2-mediated H3K27me3 specifically suppresses MyD88-dependent pro-inflammatory responses in macrophages. The absence of EZH2 causes deficient LPS- and Pam3CSK4-induced activation of NF-κB through MyD88-dependent TLR rather than poly (I:C)-triggered TLR3 stimulation [105]. In contrast, EZH2 is dispensable in the MyD88-dependent TLR-induced activation of MAPK. Zhang et al. have discovered that EZH2-activated suppression of cytokine signaling 3 (Socs3), an anti-inflammatory gene and an inhibitor of JAK/STAT signaling, mediates the ubiquitination of TRAF6 [105,106]. Consistently, the deficiency of EZH2 markedly alters the degradation of TRAF6, repressing the subsequent activation of NF-κB. All the above studies indicate that EZH2 can be developed as a target for immunotherapy.

## 4. Summary and Perspectives

Conserved methyltransferases and methyltransferase-binding proteins are important in the development and reprogramming of cells, tissues, and organs of the immune system. Recently, there has been a dramatic proliferation of reports on these important epigenetic regulators. Given the importance of lysine and arginine methylation as regulators in a variety of immunological diseases, targeting methyltransferases has gained much attention as a therapeutic strategy. This strategy requires greater insight into the structure and mechanism of methyltransferases as well as information on the molecular complex formation with their biding proteins. Pharmacological inhibitors or agonist/antagonist of methyltransferases could be developed as a novel therapeutic strategy against immunological diseases, which warrants further investigation. In particular, more knowledge at the molecular docking levels between methyltransferases and their counterpart substrates can lead to the development of specifically targeted drugs against the binding between a methyltransferase and a substrate in pathophysiological response in immunological disease conditions.

## Figures and Tables

**Figure 1 molecules-29-00360-f001:**
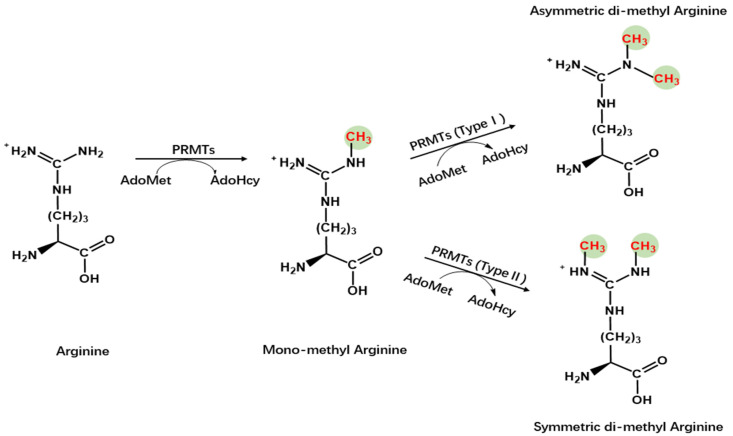
Chemistry of arginine methylation. Type I protein arginine methyltransferases catalyze asymmetric methylation of arginine, whereas Type II protein arginine methyltransferases catalyze symmetric demethylation in arginine.

**Figure 2 molecules-29-00360-f002:**
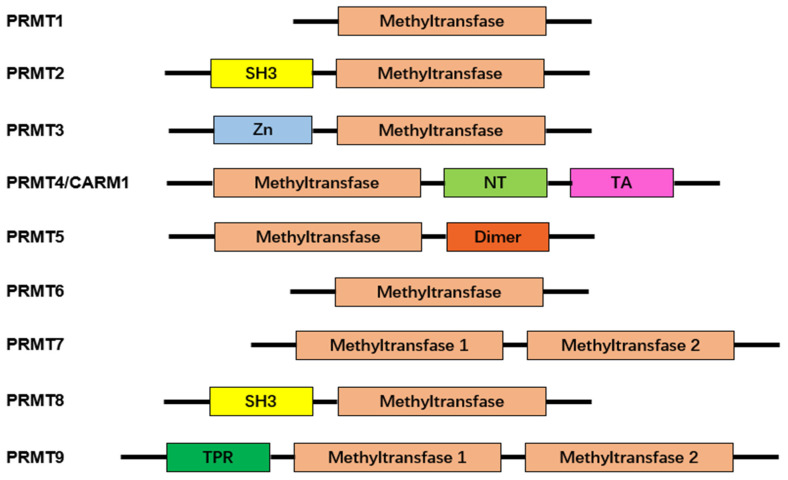
Characteristic domains of arginine protein methyltransferases. Arginine protein methyltransferases (PRMTs) share a common structural domain, the methyltransferase domain. Additionally, some members of PRMTs feature distinct structural domains, enhancing their functional diversity. These additional domains include Src-homology 3 (SH3), zinc-finger (Zn), nuclear translocation (NT), transactivation (TA), dimerization (Dimer), and tetratricopeptide repeat (TPR). The presence of these characteristic domains contributes to the unique roles and regulatory functions exhibited by different PRMTs.

**Figure 3 molecules-29-00360-f003:**
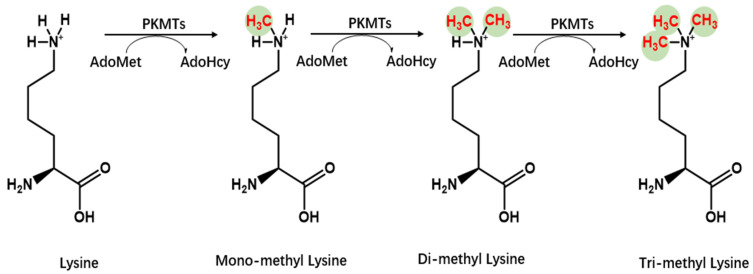
Chemistry of lysine methylation. Lysine methyltransferases (KMTs) facilitate the addition of methyl groups to substrates. Typically, the lysine ε-amino groups can accommodate up to three methyl groups, leading to the formation of mono-, di-, or trimethyllysine.

**Figure 4 molecules-29-00360-f004:**
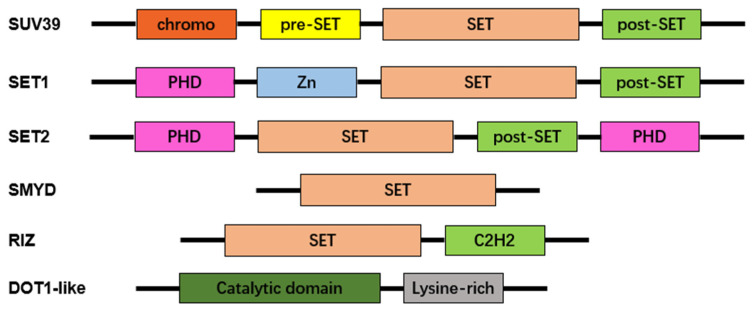
Characteristic domains of lysine protein methyltransferases. Most lysine methyltransferase (KMT) families exhibit an evolutionarily conserved SET domain, except for the DOT1-like family. Additional domains include chromo shadow (chromo), plant homeodomain finger (PHD), zinc-finger (Zn), and Cys2–His2 zinc finger (C2H2) domains.

**Figure 5 molecules-29-00360-f005:**
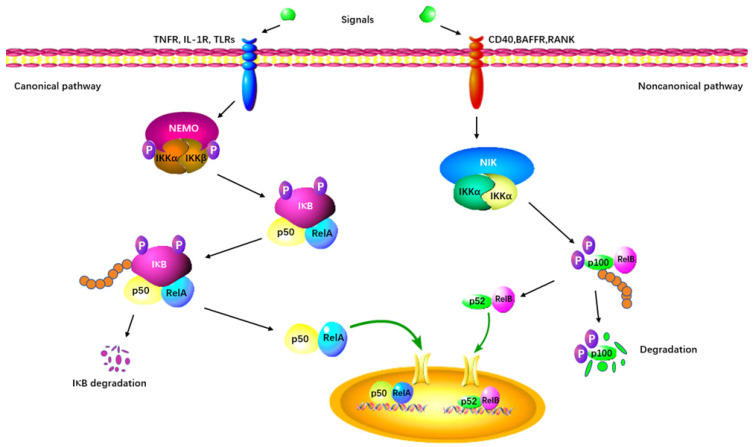
The canonical and non-canonical NF-κB signaling pathway. The canonical pathway is triggered by TLRs, TNFRs, and IL-1R, leading to the phosphorylation and degradation of the inhibitory protein IκB. NF-κB is activated by its release from the IκB-containing complex and translocates into the nucleus. The non-canonical pathway depends on the activation of p100/RelB complex by BAFFR, CD40, and RANK. This cascade involves the phosphorylation of NIK, which in turn phosphorylates IKKα. Subsequently, the p52-RelB heterodimer is activated and translocates to the nucleus.

**Figure 6 molecules-29-00360-f006:**
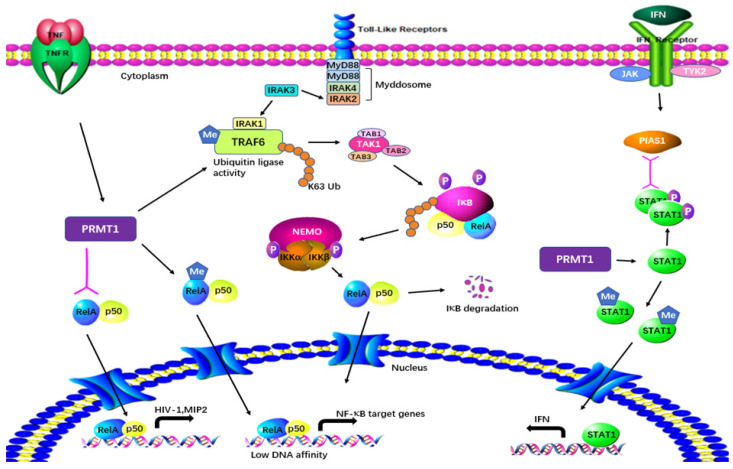
Regulatory mechanisms of PRMT1 in inflammatory responses. PRMT1 plays a multifaceted role in cellular processes by directly binding to and methylating TRAF6. The arginine methylation of TRAF6 by PRMT1 downregulates its ubiquitin ligase activity, resulting in the suppression of basal NF-κB activation. PRMT1 also methylates RelA, regulating DNA affinity and the expression of NF-κB target genes. The interaction between PRMT1 and RelA activates NF-κB-dependent gene expression at the promoters of MIP2 and HIV. Furthermore, PRMT1 methylates PIAS1, influencing IFN transcription. Additionally, PRMT1 methylates STAT1 on Arg-31, and this methylation is essential for transcriptional activation.

**Figure 7 molecules-29-00360-f007:**
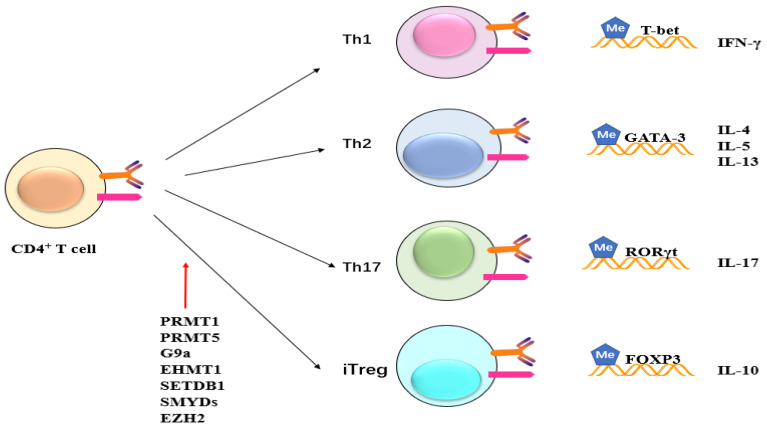
Potential role of methylation on CD4^+^ cells. Methyltransferase like PRMT1, PRMT5, G9a, EHMT1, SETDB1, SMYDs, and EZH2 exert their influence on CD4^+^ T cells, impacting subsequent T cell differentiation and maturation. This, in turn, leads to alterations in the expression of inflammatory factors such as IFN and ILs.

**Table 1 molecules-29-00360-t001:** Regulatory mechanisms of protein arginine methyltransferases (PRMTs) in immunity.

PRMTs	Pathology/Affected Field	Treatment/Model	Target Molecules (Binding Partner)	Observations	Refs.
PRMT1	Toll-like receptor signaling	Primary and cultured cells	TRAF6	-Decreased ubiquitin ligase activity of TRAF6 -Reduced activation of Toll-like receptor signaling -Suppressed basal NF-κB activation	[29]
	Macrophages	PRMT1 mutation PARP1^−/−^ macrophages	p65 (p50), PARP1	-Activated NF-κB-dependent gene expression	[30]
	Inflammatory and immune responses	Knockdown of PRMT1	RelA, p65 (p50)	-Increased levels of NF-κB target genes	[31]
	IFN-dependent responses	Methyltransferase inhibitor	PIAS1	-Decreased anti-viral and anti-proliferative abilities of type I interferons	[32,33,34]
	Innate immune responses	Myeloid-specific PRMT1 knockout mice	PPARγ	-Caused a lower survival rate and higher pro-inflammatory cytokine production	[35]
	Humoral immunity in B cells	PRMT1-impaired B cells		-Decreased the immune system response to T cell-dependent antigens -Reduced survival, proliferation, and differentiation of B cells	[36]
	T cells	NIP45-impaired mice	NIP45 (NFAT)	-Deficient expression of IFN-γ and IL-4	[37]
	Th17 cells	Knockdown of PRMT1 by shRNA, specific PRMT1 inhibitor autoimmune encephalomyelitis in mice	RORγt	-Regulated the production of Th17 cells and Th17 differentiation -Alleviated activation of EAE in mice	[38]
	Human and mouse T cells	Transmethylation inhibition	Vav1 (Rac)	-Reduced methylation of Vav1 and IL-2 production	[39,40]
CARM1	Immune responses	CARM1^−/−^ mouse embryonic fibroblasts	p160 (ER), p300 (BRCA1), p65	-Dampened expression of a group of NF-κB target genes	[41,42,43]
	Macrophages	LPS stimulation	RNA-binding protein HuR	-Stabilized TNF-α mRNA	[44]
	Thymocytes	CARM1-deleted embryos		-Reduced the number of thymocytes	[45]
PRMT5	T cell-mediated immune dysfunction	aGVHD mouse model, inhibitor of PRMT5	ERK1/2, STAT1	-Improved survival and reduced disease incidence and clinical severity -Decreased phosphorylation of STAT1 and ERK1/2 and transcription of pro-inflammatory genes	[46]
	T cells	Autoimmune encephalomyelitis (EAE) mouse model inhibitor of PRMT5		-Repressed memory T cell responses -Downregulated IL-2 production and proliferation of recall Th cells	[47]
	Lymphoma cells	PRMT5 knockdown by shRNA		-Regulated TP53K372 methylation, cyclin D1 transcriptional activation, BCL3 production	[48]
	Tregs	Conditional knockout of PRMT5 mice, pharmacological inhibition	FOXP3	-Developed severe scurfy-like autoimmunity -Reduced human Treg functions	[49]
	Natural killer T cells	T cell-specific PRMT5 conditional knockout mice		-Led to peripheral T cell lymphopenia in mice -Impaired IL-7-mediated survival and TCR-induced proliferation in vitro	[50]
	Pro-B and pre-B cells	Conditional deletion of PRMT5 in pro-B cells		-Severe deficit in antibody-secreting cells -Reduced pre-immune serum IgG1	[51]
PRMT6	Tumor-associated macrophages	Tamoxifen-inducible lung-targeted PRMT6 gain-of-function mouse model	ILF2	-Regulated pro-inflammatory genes: TNFα and iNOS	[52]
	HIV	Knockdown of PRMT6	HIV-1 Tat	-Enhanced HIV-1 production and faster viral replication	[53]
	Inflammatory responses	Transgenic mice that ubiquitously express PRMT6 fused to the hormone-binding portion of the estrogen receptor	RelA	-Regulated NF-κB target genes	[54]
	Anti-viral innate immunity	PRMT6-deficient mice	IRF3	-Promoted the TBK1–IRF3 interaction -Enhanced IRF3 activation and type I interferon production	[55]

**Table 2 molecules-29-00360-t002:** Regulatory mechanisms of protein lysine methyltransferases (PKMTs) in immunity.

KMTs	Pathology/Affected Field	Treatment/Model	Target Molecules (Biding Partners)	Observations and Molecular Effects	Refs.
G9a	Macrophages	ATF7^−/−^ peritoneal macrophages	ATF7 (M-phase phosphoprotein 8)	-Larger subpopulation of cells expressing MHC class II and CD86	[75]
	RNA virus infection	G9a^−/−^ flies		-More sensitive to RNA virus and succumb faster to infection, activated JAK/STAT pathway	[76]
	CD4^+^ T helper cells	G9a-deficient mouse model		-Impaired induction of IL-4, IL-5, and IL13, mediated Th cell differentiation	[77]
	Intestinal inflammation	G9a^−/−^ T cells		-Increased frequency of IL-17A-producing and Foxp3^+^ Tregs	[78]
		Methyltransferase activity inhibitors		-Promoted Th17 and Treg differentiation	[78]
	Acute viral infection	Blimp-1-deficient CD8^+^ T cells	Blimp-1 (Groucho, H3)	-Activated CD25 and CD27, enhanced proliferation and survival	[79]
	Innate lymphoid cells	Mice with a hematopoietic cell-specific deletion of G9a		-Deficient development of immature ILC2s -Greater resistance to allergic lung inflammation	[80]
EHMT1	Viral infection	EHMT1-specific siRNAs	p50 (p65)	-Promoted IL-8 production in response to TNFα stimulation	[81,82]
	CD4^+^ Treg cells	Knockout of EHMT1 in primary cells	Wiz	-Enhanced Foxp3 expression during iTreg differentiation	[83]
	Kaposi’s sarcoma-associated herpesvirus	Knockdown of EHMT1 by shRNA	IFI16	-Regulated KSHV lytic genes	[84]
SETDB1	Macrophages	Knockdown of SETDB1 by shRNA		-Augmented TLR4-mediated NF-κB recruitment -Promoted interleukin-6 promoter activity.	[85]
		Macrophage-specific SETDB1-knockout mice		-Higher serum interleukin-6 concentration in response to LPS -More sensitive to endotoxin shock	
	Autoimmune diseases	Experimental autoimmune encephalomyelitis (EAE)	OX40	-Regulated IL-17 expression	[86,87]
		Knockdown of SETDB1 by shRNA			
	T cell development	SETDB1-knockout T cells	FcγRIIb	-Decreased the numbers of CD69^+^ and T-cell receptor TCRβ^+^ cells -Promoted apoptosis in the double-positive compartment	[88,89]
	B cell development	Deficient SETDB1 in pro-B cells		-Diminished B cell development	[90,91]
	Innate immunity	Knockout and knockdown of SETDB1		-Induced type I interferon response	[92]
SMYDs	Macrophages	SMYD2-specific siRNA1		-Regulated expression of pro-inflammatory cytokines and MHC-II -Regulated NF-κB and ERK signaling	[93]
	Treg cells	SMYD3^−/−^ mice		-Increased severity of pulmonary viral infection	[94]
	Macrophages	siRNAs directed against SMYD5		-Elevated transcriptional responses of a group of TLR4 target genes	[95]
EZH2	Adaptive lymphocytes	Murine model of lymphoid-specific EZH2 deficiency		-Promoted Cdkn2a and generation of IFN-γ and IL-10 under stimulation	[96]
	T cells	EZH2-deficient Th cells		-Generated higher levels of IL-4, IL5, IL-13, and IFN-γ during respective stimulation	[97,98]
		DO11.10/Rag2^−/−^ transgenic mice		-Enhanced production of T-bet and Gata-3	[99]
	CD4^+^ T cells	EZH2^fl/fl^ mice		-Reduced numbers of Treg cells in vivo and differentiation in vitro, enhanced the production of memory CD4 T cells	[100]
	Effector T cells	EZH2^fl/fl^ mice		-Accelerated effector Th cell death	[101]
	Pro-B cells, pre-B cells	EZH2^fl/fl^ mice		-Reduced the frequencies and numbers of pre-B cells and immature B cells	[102]
	Germinal center B-cells	Conditional EZH2 knockout and EZH2Y641N knockin mice Methyltransferase activity inhibitors		-Decreased the number of splenic GC B cells -Defected affinity maturation of immunoglobulin and formation of high-affinity antibodies	[103]
	B cells	Methyltransferase activity inhibitors		-Reduced the proliferation of pre-B cells and B cells	[104]
	Macrophages	Myeloid cell-conditional EZH2 knockout mice		-Attenuated autoimmune inflammation -Stimulated Socs3 -Enhanced degradation of TRAF6	[105,106]

## Data Availability

Not applicable.

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
