# Peer review of "The Role of Protein Methyltransferases in Immunity"

_molecules, 2024, doi:10.3390/molecules29020360_

Round 1

Reviewer 1 Report

Comments and Suggestions for Authors

Song and collaborators describe in the review "The role of protein methyltransferases in Immunity" the importance of the effect of methyltransferases, the type and degree of methylation on immunity, to understand the mechanism of this process.

In order to publish the manuscript, you must attend to the following observations.

In all figures, it is necessary to complete the figure caption, not just write the title of the figure; it must include text that explains and specifies the most relevant part of the image. Be careful not to repeat what is included in the manuscript's text.

Review the grammar of the language.

As it is a review, it is essential to refer to the sources from which the information is consulted. References are missing in several paragraphs. Likewise, to correct the sequence of references, it is recommended to use a bibliographic manager, such as Zotero. For example, on line 132, reference 37 is referred and then 63. The entire text must be reviewed.

It is essential to update references; more recent references must be included.

 Line 29: what does SET mean? Put the full name since it is mentioned for the first time in the text.

Line 144. Writing "et al." in italics is essential since it is in Latin.

Include a summary and prospect section in which the authors' opinions are included and the most outstanding aspects are recapitulated.

Comments on the Quality of English Language

The correctness of the grammar must be attended to; it contains moderate errors.

Author Response

Reviewer #1

Song and collaborators describe in the review "The role of protein methyltransferases in Immunity" the importance of the effect of methyltransferases, the type and degree of methylation on immunity, to understand the mechanism of this process.

In order to publish the manuscript, you must attend to the following observations.

In all figures, it is necessary to complete the figure caption, not just write the title of the figure; it must include text that explains and specifies the most relevant part of the image.

#: We have included more sentences in each figure caption (see L51-53, 72-78, 97-100, 114-118, 105-144 and 209-217).

Be careful not to repeat what is included in the manuscript's text.

#: We have thoroughly checked again and revised it.

Review the grammar of the language.

#: We have firstly edited it by E-World editing company and we will use editing service from MDPI publishing. Otherwise, we will get additional language editing service from E-World editing company if you give use 10 days more.

As it is a review, it is essential to refer to the sources from which the information is consulted.

References are missing in several paragraphs. Likewise, to correct the sequence of references, it is recommended to use a bibliographic manager, such as Zotero. For example, on line 132, reference 37 is referred and then 63. The entire text must be reviewed.

#: We apologize for any inconvenience. This was due to the citation in Tables. The entire article has been reviewed, and the identified issues have been addressed.

It is essential to update references; more recent references must be included.

#: We appreciate your suggestion. The references in the introduction have been revised accordingly (L44, 84, 85, 90, Table 1, 205, 323, 327, 331, 439, 446, and 487).

 Line 29: what does SET mean? Put the full name since it is mentioned for the first time in the text.  

#: Thank you for bringing this to our attention; it was an oversight on our part. The meaning of SET has already been explained where it first appeared (see L30-32, SET: Su(var)3-9, Enhancer-of-zeste and Trithorax).

Line 144. Writing "et al." in italics is essential since it is in Latin.

#: Line 148, "et al." has been changed to italics and others have been also corrected (see L153, 161, 176, 251, 275, 324, 345, 409, 425, 434, 445, 472, 487, and 517).

Include a summary and prospect section in which the authors' opinions are included and the most outstanding aspects are recapitulated.

#: We have added this point in Section 4 Summary and Perspectives section (see L529, 531-532, and 533-538)

Reviewer 2 Report

Comments and Suggestions for Authors

The authors of the paper entitled “The role of protein methyltransferases in immunity” collected data on the role of methyltransferases in many immune stages. The authors have experience in the topic confirmed by their previous publications in the field. The novelty of the paper is good, the topic is interesting and the data provided useful information. The article has the typical construction of a review article. The diagrams and tables are clear and represent well the relationships described. 

However, I have a few comments about the article:

1)      Keywords : word „immunity” I suggest replace the word or deleted it so that it does not coincide with the words in the title

2)      Line 53- abbreviation "SET” should by explained, line 240 is not the first time when “SET“ appears

3)      I suggest to add references in lines 262, 372, 383, 399.

4)      Line 421 : [104] should not be written in italics

Summarizing, the article is interesting, brings new knowledge to the topic and may be published in the Molecules after minor revision.

Author Response

Reviewer #2

The authors of the paper entitled “The role of protein methyltransferases in immunity” collected data on the role of methyltransferases in many immune stages. The authors have experience in the topic confirmed by their previous publications in the field. The novelty of the paper is good, the topic is interesting and the data provided useful information. The article has the typical construction of a review article. The diagrams and tables are clear and represent well the relationships described. 

However, I have a few comments about the article:

1)      Keywords : word „immunity” I suggest replace the word or deleted it so that it does not coincide with the words in the title

##: Thanks to your careful advice, “immunity” in key words has been removed.

2)      Line 53- abbreviation "SET” should by explained, line 240 is not the first time when “SET“ appears

##: Thank you for bringing this to our attention; it was an oversight on our part. The meaning of SET has already been explained where it is first appeared (see L30-32, SET: Su(var)3-9, Enhancer-of-zeste and Trithorax).

3)      I suggest to add references in lines 262, 372, 383, 399.

##: Yes, we agree with this comment and the relevant literature has been added (see L323, 327, 331, 333, 439, 446, 449, and 466).

4)      Line 421 : [104] should not be written in italics

##:  The format has been corrected. Thanks a lot (see L488).

Summarizing, the article is interesting, brings new knowledge to the topic and may be published in the Molecules after minor revision.

##: Thanks very much for your good words.

Reviewer 3 Report

Comments and Suggestions for Authors

Protein methyltransferases (PMTs) comprise a major class of epigenetic regulatory enzymes that have an important role in the regulation of gene expression. Additionally, these proteins have roles in signal transduction, RNA-binding proteins, ribosome biogenesis, and splicing. In this manuscript, Song and collaborators make an extensive review of the arginine and lysine methyltransferases and their role in immune response modulation.

The study of these enzymes is relevant because of their important effects on the regulation of gene expression as well as the regulation of the expression of important cellular processes.

Nevertheless, the volume of material presented in the manuscript makes it difficult to read, and the topics they choose to cover in order to explain the various PRMT and KMT kinds are inconsistent.

The authors explain the various domains and motifs of the proteins; however, a figure displaying these proteins' diagrams or their three-dimensional structure would be helpful for improved understanding.

The authors do not clearly explain the molecular effect of these modifications, concerning which genes are regulated in response to histone methylation and whether there is activation or repression, as well as emphasizing protein-protein interactions between methylated proteins.

The authors should take into account the concepts of writer, eraser, and readers about these enzymes, which would make the text more topical.

It would be preferable to concentrate the tables on biological processes associated with the immune response, such as inflammation, virus infection, T or B cell activation, etc., as the tables are difficult to understand. Because diseases, cells, and mixed processes are present.

The figures should show the relationship between enzymes and biological processes; for example, in Figure 1, the effect of PRMTs and the NFKB factor is not observed. The figures require captions where the processes are clearly explained.

The text presents textual similarities with some of the articles.

Examples

Lines 207 to 208

Lines 222 to 224

Lines 239 to 242

Author Response

Reviewer #3

Protein methyltransferases (PMTs) comprise a major class of epigenetic regulatory enzymes that have an important role in the regulation of gene expression. Additionally, these proteins have roles in signal transduction, RNA-binding proteins, ribosome biogenesis, and splicing. In this manuscript, Song and collaborators make an extensive review of the arginine and lysine methyltransferases and their role in immune response modulation. The study of these enzymes is relevant because of their important effects on the regulation of gene expression as well as the regulation of the expression of important cellular processes. Nevertheless, the volume of material presented in the manuscript makes it difficult to read, and the topics they choose to cover in order to explain the various PRMT and KMT kinds are inconsistent.

###: Thanks very much for your comments. We have tried to improve the quality of our paper by introducing some points raised by the reviewer. Please see below.

The authors explain the various domains and motifs of the proteins; however, a figure displaying these proteins' diagrams or their three-dimensional structure would be helpful for improved understanding.

###: Thanks to your valuable suggestion, we show the structure and schematic of PRMTs and PKMTs in Figures 2 and 4, respectively.

The authors do not clearly explain the molecular effect of these modifications, concerning which genes are regulated in response to histone methylation and whether there is activation or repression, as well as emphasizing protein-protein interactions between methylated proteins.

###: Thanks for your advice. Tables 1 and 2 provide a summary of the target molecule of the methyltransferase (if reported in the literature), the treatment or model used in the literature (e.g., knockout mouse), binding partners, and the effects caused by that treatment, like decreased ubiquitin ligase activity of TRAF6, activated NF-κB-dependent gene expression (see Table 1 and 2).

The authors should take into account the concepts of writer, eraser, and readers about these enzymes, which would make the text more topical.

###: Thank you for your suggestion. We have added these concepts to the introduction (Line 23-26).

It would be preferable to concentrate the tables on biological processes associated with the immune response, such as inflammation, virus infection, T or B cell activation, etc., as the tables are difficult to understand. Because diseases, cells, and mixed processes are present.

###: We agree with your suggestion. Regulated process like post-translation modification have been removed from the form as they make the form look confusing (see Tables 1 and 2).

The figures should show the relationship between enzymes and biological processes; for example, in Figure 1, the effect of PRMTs and the NFKB factor is not observed. The figures require captions where the processes are clearly explained.

###: We apologize for the unclear layout of this essay. The role of PRMTs and PKMTs is explained in the Tables 1 and 2, and Figures 6, and 7. Figure 1 illustrates only the chemical processes related to methylation by PRMTs.

The text presents textual similarities with some of the articles.

Examples

Lines 207 to 208

###: We rewrote the part you mentioned to avoid similarities with other papers (see L272-275).

Lines 222 to 224

###: We rewrote the part you mentioned to avoid similarities with other papers (see L288-293).

Lines 239 to 242

###: We rewrote the part you mentioned to avoid similarities with other papers (see L307-310).

Round 2

Reviewer 3 Report

Comments and Suggestions for Authors

I had reviewed the revised version of your manuscript, entitled The Role of Protein Methyltransferases in Immunity, and I noticed that all the suggestions provided during the initial review had been carefully addressed. The manuscript is now much clearer, more coherent, and of higher quality overall.

Author Response

Thanks very much for your good words.